# Gene Editing for Plant Resistance to Abiotic Factors: A Systematic Review

**DOI:** 10.3390/plants12020305

**Published:** 2023-01-09

**Authors:** Fernanda dos Santos Nascimento, Anelita de Jesus Rocha, Julianna Matos da Silva Soares, Marcelly Santana Mascarenhas, Mileide dos Santos Ferreira, Lucymeire Souza Morais Lino, Andresa Priscila de Souza Ramos, Leandro Eugenio Cardamone Diniz, Tiago Antônio de Oliveira Mendes, Claudia Fortes Ferreira, Janay Almeida dos Santos-Serejo, Edson Perito Amorim

**Affiliations:** 1Department of Biological Sciences, Feira de Santana State University, Feira de Santana 44036-900, BA, Brazil; 2Embrapa Mandioca e Fruticultura, Cruz das Almas 44380-000, BA, Brazil; 3Embrapa Soja, Londrina 86000-000, PR, Brazil; 4Department of Biochemistry and Molecular Biology, Federal University of Viçosa, Viçosa 36570-000, MG, Brazil

**Keywords:** CRISPR/Cas9, CRISPR/Cas12a, genome editing, abiotic stresses, genetic improvement, tolerance, state of the art

## Abstract

Agricultural crops are exposed to various abiotic stresses, such as salinity, water deficits, temperature extremes, floods, radiation, and metal toxicity. To overcome these challenges, breeding programs seek to improve methods and techniques. Gene editing by Clustered Regularly Interspaced Short Palindromic Repeats—CRISPR/Cas—is a versatile tool for editing in all layers of the central dogma with focus on the development of cultivars of plants resistant or tolerant to multiple biotic or abiotic stresses. This systematic review (SR) brings new contributions to the study of the use of CRISPR/Cas in gene editing for tolerance to abiotic stress in plants. Articles deposited in different electronic databases, using a search *string* and predefined inclusion and exclusion criteria, were evaluated. This SR demonstrates that the CRISPR/Cas system has been applied to several plant species to promote tolerance to the main abiotic stresses. Among the most studied crops are rice and *Arabidopsis thaliana*, an important staple food for the population, and a model plant in genetics/biotechnology, respectively, and more recently tomato, whose number of studies has increased since 2021. Most studies were conducted in Asia, specifically in China. The Cas9 enzyme is used in most articles, and only Cas12a is used as an additional gene editing tool in plants. Ribonucleoproteins (RNPs) have emerged as a DNA-free strategy for genome editing without exogenous DNA. This SR also identifies several genes edited by CRISPR/Cas, and it also shows that plant responses to stress factors are mediated by many complex-signaling pathways. In addition, the quality of the articles included in this SR was validated by a risk of bias analysis. The information gathered in this SR helps to understand the current state of CRISPR/Cas in the editing of genes and noncoding sequences, which plays a key role in the regulation of various biological processes and the tolerance to multiple abiotic stresses, with potential for use in plant genetic improvement programs.

## 1. Introduction

The assertion that increases in population growth demand the production of more food is now a reality. According to data from the Food and Agriculture Organization, in 2050, the world population will reach 10 billion, an increase of approximately 30% when considering data from 2020, and a fact that reinforces the need for greater food supply in the coming years [1,2]. In addition, the effects of climate change will exacerbate the damage caused by pests and diseases to different agricultural crops, drastically reducing productivity [3].

Agriculture has always been negatively influenced by different abiotic stresses which have the potential to reduce average productivity by up to 50%, mainly by salinity, water deficit, temperature extremes, floods, radiation, ionic toxicity, and metal toxicity [4,5]. These limiting factors have led to the constant search for improvements in agricultural crops, especially with regard to productivity and quality of seeds, grains, and fruits, in addition to resistance to stress.

Although genetic improvement remains an effective and long-lasting technique for improving crops, it still faces some challenges, such as the complex inheritance of the vast majority of agronomic traits and strong genotype–environment interaction [6]. Breeding programs seek to reinforce their techniques through new technologies and knowledge. Approaches that combine genetic engineering and *omics* technologies are promising because they allow studies directly on a genotype and its relationship with the phenotype. However, there are limitations, including ethical issues, including the safety and efficacy of new technologies and the possible impact on the environment, such as cross-pollination between genetically edited and non-edited plants, that can lead to plants resistant to herbicides and possible resistance to pests and pesticides [7,8,9].

In recent years, gene editing based on a natural immune system used by bacteria to prevent infection by viruses has shown promise [10,11,12]. The clustered regularly interspaced short palindromic repeats (CRISPR)/CRISPR-associated protein (CRISPR/Cas), derived from the adaptive immunity system of *Streptococcus pyogenes*, is used in genetic engineering for the precise modification of organismal DNA and RNA and is considered an easy-to-use and cheaper method compared to other existing ones. In addition, it has the potential to change genetic improvement strategies in a revolutionary way, since CRISPR/Cas tools offer modifications in all layers of the central dogma of molecular biology, with epigenetic, transcriptional, translational and post-translational modifications of proteins [13,14,15,16].

Editing the genomes of plants using CRISPR is achieved through three fundamental steps: (i) a projected sequence-specific nuclease (SSN) and endonuclease CRISPR/(Cas9, Cas12a or others), which induces a double-strand break (DSB) at a target DNA site; (ii) the DSB at the target site can be repaired by any of the non-homologous end joining (NHEJ) pathways or error-free homology-directed repair (HDR); and (iii) afterwards, the NHEJ pathway creates small insertions or deletions of nucleotides (*indels*) at the target site which leads to frame-shift mutations while precise DNA repair occurs in the case of HDR through the introduction of a donor strand of a desired sequence [17,18,19].

After the discovery of CRISPR/Cas, several tools based on this system were developed, allowing editing of target genomic loci and beyond [20]. Such tools include DNA base editors (BEs), epigenetic modifiers, prime editors (PEs), and transcription regulators (CRISPRa and CRISPRi) [21]. Several studies on CRISPR/Cas9 in plants have been conducted, both in monocotyledons and dicotyledons related to different abiotic stresses, such as osmotic stress in *Arabidopsis thaliana* [22], salinity in tomato [23], and water deficit in wheat [13], rice [24], chickpea [25], and corn [26], as well as temperature extremes in soybean and grapevine [27,28].

Studies regarding gene editing via CRISPR/Cas9 in agricultural crops with a focus on abiotic stresses are discussed in reviews of the literature [8,19,29,30,31,32,33,34,35,36], highlighting systematic reviews (SR) that focused on the occurrence of off-targets and motifs implied in this phenomena [37,38]. Our SR shows the role of gene editing as an auxiliary tool in genetic breeding programs aiming to develop more adapted cultivars to different abiotic factors, to date. This SR aims to provide a comprehensive and impartial compilation of many relevant studies in a single document through systematized searches in electronic databases. This approach is widely used in research in fields related to humans, especially in those related to identifying information on drug efficiency, the adverse effects of drugs, the mechanisms of action of drugs, etc. [39,40,41,42]. For this approach, systematic and defined methods are used a priori in the identification and selection of studies, data extraction, and analysis of results [43,44,45,46]. Thus, this SR involves the identification and comparison of data from the last seven years on the use of CRISPR/Cas technology in the editing of genes for tolerance to abiotic stresses.

## 2. Results

### 2.1. Research in Electronic Databases

Initially, 3597 articles were identified in the electronic databases (Figure 1). Central PubMed had the largest number of articles (1140), followed by Google Scholar (990) and the CAPES journals portal (796). Springer, Web of Science, and CABI direct databases accounted for 248, 217, and 193 articles, respectively. In addition, seven potentially relevant studies published after the selection processes were added manually (Figure 1).

A total of 1312 duplicate articles were excluded, and 1969 were excluded at the selection stage after reading the abstracts and keywords. During the extraction step, 310 studies were analyzed, and after reading the articles in full, 175 were excluded because they did not meet the inclusion criteria. A total of 141 articles were accepted for the SR, of which 119 were experimental articles and 22 were literature reviews (Figure 1). For reference purposes, the manuscripts were stored in an open-access digital library at https://doi.org/10.5281/zenodo.7413836 (accessed on 9 December 2022).

A bibliometric map was generated from the frequency of keywords in the experimental articles (*n* = 119) (Figure 2). This SR sought to find articles on CRISPR/Cas being used to edit genes for tolerance to abiotic stresses published in the last seven years (2015–2022); however, no studies were found prior to 2016. The frequency of generated studies was higher during 2019 and 2020 than during other years (Figure 2).

The terms CRISPR or CRISPR/Cas9 were more frequent in articles related largely to water and salt stress; in addition, they were also related to the terms abscisic acid or ABA and Arabidopsis, rice, and tomatoes.

To identify which scientific journals published the most articles on CRISPR/Cas in terms of editing genes related to abiotic stresses in plants, a word cloud was generated (Figure 3). The most frequent journal was Frontiers in Plant Science, followed by BMC Plant Biology and Plant Physiology. Other journals, such as the International Journal of Molecular Sciences, Plant Molecular Biology, Plant Biotechnology Journal, Journal of Experimental Botany, Plant Science, and Plants, had notable frequencies on the map (Figure 3).

### 2.2. Study Sites

The studies were mostly conducted in Asia, with 95 studies occurring in China, with rice (50) being the most studied crop (Figure 4). Countries such as South Korea (4), Japan (3), and Russia (2) were also producers of knowledge about the use of the CRISPR/Cas system in plant breeding for tolerance to abiotic stresses. On the American continent, the United States of America had the highest frequency of articles (5), and the *Arabidopsis thaliana* model plant was the most studied in this country.

### 2.3. Edited Plant Crops and Abiotic Stresses

According to FAOSTAT (2022), twenty main agricultural crops were considered to create a string search. Four model plants were included, *Arabidopsis thaliana*, *Nicotiana tabacum*, rice, and corn. Although 24 plants were inserted into the string data of our review, based on FAOSTAT data, only 15 involved studies with CRISPR/Cas for abiotic factors.

Among the crops edited using CRISPR/Cas, rice was the most studied, with 50 articles and the most reported abiotic stresses were: salinity (18), water deficit (16), salinity and water deficit (9), and tolerance to cold (7) (Figure 5). *A. thaliana*, considered the model plant in biological experiments, was the second most studied plant in this SR (24), with publications on low temperatures (6) and water deficit prevailing (4). For tomato (17), corn (7), and soybean (7), the highest frequency of studies related to water deficit tolerance and salinity. The other topics with low article frequencies focused on stress caused by high temperatures, water deficit, salinity, and osmotic stress (Figure 5).

### 2.4. Types of Explants

Regarding the most commonly used explants for gene editing, for rice, embryogenic calli (26) had the highest number of articles. For *A. thaliana*, tomato, corn, and soybean, inflorescences (12), cotyledons (11), embryos (6), and seeds (4) were most commonly used as explants, respectively (Figure 6).

### 2.5. Main Genes and Metabolic Pathways

The word cloud shown in Figure 7 shows the frequency of genes that appear in studies on CRISPR/Cas technology used to edit genes for tolerance to abiotic stresses. Tolerance-related genes included *C-repeat/DRE-Binding Factor* (*CBFs 1, 2* and *3*), *9-cis-epoxycarotenoid dioxygenase* (*NCDE3*), tomato gene *Auxin Response Factors 4* (*SlARF4*), *Mitogen-activated protein kinases* (*SlMAPK3*), and *microRNAs* (*miRNAs*) related to cold tolerance, multiple stresses, and tolerance to salinity and water deficit, respectively.

Among the most reported metabolic pathways in the articles, ABA biosynthesis (17) and jasmonic acid biosynthesis (2) were the most frequently cited. The other metabolic pathways were present in only one article, and 74 articles did not perform this type of evaluation (Figure 8).

### 2.6. Methods of Editing with CRISPR

The selected articles cited different protocols for gene editing, including the protocols of [47,48], which were the most used (Appendix A). All articles used endonuclease Cas9, except for one that used endonuclease Cas12a, previously named Cpf1. To target these endonucleases, the authors used single guide RNA (gRNA) in one to seven of the articles. Several vectors, such as PHEC401, pHEE-FT, pYLCRISPR, and pRGEB31, were used; however, pCAMBIA and its derivatives were the most cited. Regarding the strategy used for DNA repair, all were based on non-homologous end joining (NHEJ), with the exception of two articles that used the homology-directed repair (HDR) strategy (Appendix A). Appendix A shows the primers used to construct the vectors and gRNA.

To introduce the gene of interest into plant cells, the most commonly used delivery methods were via *Agrobacterium tumefaciens* (69), *Agrobacterium rhizogenes* (4), and ribonucleoproteins (RNPs) (2) (Figure 9A).

Off-target sites were frequently analyzed for the occurrence of off-target activity using detection methods. Among the methods used for CRISPR/Cas mutation detection, sequencing was the most used (81), followed by polymerase chain reaction (PCR) (70) (Figure 9B). Other methods, such as digestion with restriction enzymes (2), T7E1 assay (3), Inference of CRISPR Edits (ICE) (1), NCBI-blast-primer blast (1), PCR and alignment with control sequences (1), Restriction Fragment Length Polymorphism (RFLP) (1), quantitative reverse transcription PCR (RT-qPCR) (2), and Western blot (1), were also used (Figure 9B). The sequences used to detect mutations and off-targets are listed in Appendix A, and the analysis of the off-target activities, as well as the methods used and the forecasting tools, are listed in Appendix A.

### 2.7. Auxiliary Methods

Auxiliary methods used with CRISPR, i.e., those in which it was possible to confirm that CRISPR/Cas was efficient in modifying the desired characteristics in the plants, where the characteristics of the knockout were compared with a control and/or with the overexpressed mutant, were identified in the articles. Among the most used tools, reverse transcription-PCR (RT-qPCR) (109) and transgenic or cisgenic (67) analysis had higher frequencies, followed by RNA-Seq (21) and Western blotting (12) (Figure 10).

### 2.8. Risk of Bias Analysis

Based on three questions specifically chosen to characterize the risk of bias, 63 studies had low risk of bias, and 56 had moderate risk. Six studies answered “no” to question one, which was related to off-target activity. Only one article did not answer question two, which was related to phenotypic analysis, and 48 articles answered “no” to question three, which was related to evaluating the protein identified in the study. Articles that studied *miRNAs* or proteins that had already been studied were classified as “does not apply (NA)”. More information on the risk of bias assessment is in Figure 11.

### 2.9. Literature Reviews

When selecting and extracting articles for the preparation of this SR, a considerable number of literature reviews on CRISPR/Cas technology used for plant breeding with emphasis on abiotic stresses were found. It is important to highlight that only traditional literature reviews were found regarding CRISPR/Cas associated with biotic and abiotic stress, which justifies the development of a SR that considers well-defined inclusion and exclusion criteria for including articles, in contrast to a traditional literature review, which does not have well-predefined standards.

In total, 22 reviews were involved in the review with four citing specific crops, such as rice, banana, and *Arabidopsis*, and 18 focusing on abiotic stresses in different crops (Table 1). Some of the selected review articles (15) addressed different types of stresses (water deficit, high temperatures, salinity, and ionic toxicity), and all articles were more focused on using the CRISPR/Cas method to edit genes for tolerance to abiotic stresses.

## 3. Discussion

### 3.1. Research in Electronic Databases

This study systematically analyzed the literature on the use of CRISPR/Cas technology to edit genes for tolerance to abiotic stresses in plants. The RS involved searching for studies published in the last seven years (2015 to 2022); however, no studies were found prior to 2016. This may be related to the fact that the CRISPR/Cas technique is a recently developed technology, and the first studies with plants were published in August 2013 [63,64,65,66], focusing on introducing mutations into genes that would result in a distinct phenotype and would be immediately recognizable (such as the phytoene desaturase gene) to test and optimize the efficacy of the technique (‘proof of concept’) in various plant crops [67].

This SR integrated a bibliometric analysis to allow the identification of the scientific journals that publish the most studies and study keywords related to CRISPR/Cas being used to edit genes related for tolerance to abiotic stresses (Figure 2 and Figure 3). These analyses confirmed that CRISPR/Cas is the most studied technique for editing genes for tolerance to water and salt stress in *Arabidopsis* and rice in the last seven years. This data is similar to the bibliometric analysis carried out by Hamdan et al. [36] that also used the VOSviewer software and checked for the presence of terms and keywords similar to those found in our study, although this analysis was carried out with publications related to CRISPR/Cas extracted only from the SCOPUS databank in the last ten years.

### 3.2. Study Sites, Plant Crops and Abiotic Stresses

Of the 119 experimental scientific articles analyzed, 105 involved studies were conducted on the Asian continent, especially China (95), which published more on the technique than any other country. The great interest of Asian countries in studying the CRISPR system in relation to improving plant crops is probably because they are at the center of the production of several crops, with China having the highest agricultural yield in the world and accounting for a quarter world grain production [2].

Fifteen crops were reported in the scientific articles analyzed, and rice stood out with higher production in comparison to other crops on the Asian continent. This result can be explained by the fact that grain is an important staple food crop and a source of dietary supplement for more than half of the population worldwide [31]. In this RS, several abiotic stresses were studied; however, stresses due to salinity and water deficit were the most reported. Rice is grown under rainfed conditions in several agroclimatic zones and is subjected to various abiotic stresses, which trigger a series of morphophysiological and molecular responses that negatively affect growth and development and, consequently, yield potential [68,69]. Therefore, global efforts to use conventional and/or biotechnological interventions to project some response characteristics to salt stress and water deficit have occurred.

As a developing country with the largest population in the world, China is implementing efforts to employ genetic engineering technology to increase agricultural productivity, especially to improve new cultivars and agronomic practices to address the changing environment [70,71,72]. Relatedly, Cohen [73] highlights that China expanded its efforts beyond its borders in 2017, strengthening its gene editing through CRISPR/Cas, with companies specializing in crop protection and biotechnology and enabling a close relationship between government, industry, and academia.

In addition to China, countries such as South Korea (4), Japan (3), India, and the United Kingdom have published articles on rice cultivation and produced knowledge on the CRISPR/Cas system in relation to plant breeding with tolerance to abiotic stresses.

The model plant *A. thaliana* was studied in 24 articles, most of which were related to water stress and low temperatures (Figure 5). These studies used *A. thaliana* since it is the model for all plants due to its short life cycle and the feasibility of the transfer of knowledge generated to other food crops of interest [74,75]. *A. thaliana* was the first plant genome to be sequenced, and although it is well established as the model plant in plant studies, rice (*Oryza sativa*), sequenced right afterwards, seems to have more potential use as the model plant in recent studies. Besides presenting the availability of a large amount of genomic and molecular data, rice has two advantageous characteristics in the genome editing field: (*i*) it is classified as a monocotyledon and (*ii*) it is used as basic food for millions worldwide [76].

Other crops, such as tomato (17), corn (7), and soybean (7), were found at relevant frequencies. According to data from FAOSTAT [2], in the year 2020, these crops were ranked among the most produced in the world. Tomato is an important vegetable grown worldwide, and corn and soybean are also of great importance because they are consumed directly or are used in the production of animal protein, and in industries as ingredients and raw materials [77,78,79]. Water stress was the most studied stress for these three crops and for different crops in general. Water deficit is one of the most severe environmental factors limiting plant growth, development, and survival. Thus, it is necessary to develop crops tolerant to water deficits since climate change may lead to an increase in water scarcity [3]. Furthermore, plants tolerant to water deficit may meet the demand of the growing human population, which will require more food and fuel, and gene editing with the CRISPR/Cas system has emerged as an easy-to-implement system with the potential to meet this demand [80].

### 3.3. Types of Explants

Explants are small fragments of living tissue that play an important role in the efficiency of transformation and can be removed from many different parts of a plant: shoots, leaves, stems, roots, etc. [81]. A variety of explants are used in rice cultivation, with the most frequent being embryogenic calli (26) and protoplasts (6). In *A. thaliana*, tomato, corn, and soybean, inflorescences (12), cotyledons (11), embryos (6), and seeds (4) are the most commonly used explants, respectively (Figure 6).

The type of explant required for transformation differs from plant to plant. In *A. thaliana*, for example, floral immersion is a widely used method because it allows us to identify new opportunities for obtaining transgenic plants, as this approach has advantages over traditional methods, such as eliminating the need for callus cultivation in different culture media. The success and popularity of this transformation method is shown in many studies [82,83,84,85,86].

Immature embryos are the predominant explant for transformation into cereals [87,88,89,90]. Generally, immature embryos are transformed, and embryogenic calli are induced and increased under selective pressure to obtain a clonal tissue mass that can then be regenerated into plants. However, Lowe et al. [91] state that this process is laborious and time-consuming and may take from 87 to 140 days from the beginning of the transformation process to the transfer of the seedlings to the greenhouse.

Protoplasts are also used by some authors, such as Kim et al. [13], Qu et al. [92], and Zhang et al. [93]. Other studies have also reported the use of protoplasts as a preassembled ribonucleoprotein receptor explant [25,30,94]. However, the regeneration of protoplasts is still a challenge for most cultures, particularly monocotyledons [95].

### 3.4. Genes and Metabolic Pathways

A key part of the gene-editing process is the identification of target genes related to the trait for which the edit is desired, and this determines phenotypes of interest, such as tolerance to abiotic stresses. CRISPR/Cas can be used to verify gene function by direct knockout [57]. A word cloud with “genes” and “transcription factors” related to abiotic stresses used for editing by the CRISPR/Cas technique was generated (Figure 7).

There are a wide variety of regulatory genes and structural genes related to abiotic stresses, and they usually exist in the form of gene families [57]. In comparison to others, the *C-repeat/DRE* (*CBFs 1*, *2* and *3*) gene of the ligation factor had higher frequencies in the articles of this review. These genes are rapidly induced by cold stress and, in turn, activate the expression of the *Cold Regulated* (*COR*) gene, which molecularly adapts a plant to resist cold stress. Mutants were generated using the overexpression strategies RNA interference (RNAi) and CRISPR/Cas9 to characterize the *UGT79B2* and *UGT79B3* genes of the *UDP-glucosyltransferase* (*UGTs*) of *Arabidopsis*. Through these strategies, Li et al. [96] showed that *CBF1* regulates *UGT79B2/B3* and improves resistance to abiotic stress. In addition, these genes are related to freezing tolerance in many studies [82,85,97,98,99,100].

Wang et al., 2021d [101], demonstrated that light-induced cold tolerance in tomato is compromised in the *Hypocotyl3 Elongated* mutant of FAR Red (*SlFHY3*), while plants that overexpress *SlFHY3* showed higher tolerance to cold, indicating that *SlFHY3* positively regulates light-induced cold tolerance in tomato plants. The interruption of Hypocotyl5 Elongated (*SlHY5*) largely suppresses the cold tolerance of plants overexpressing *SlFHY3*, suggesting that cold tolerance mediated by SlFHY3 is dependent on *SlHY5*. The CRISPR/Cas9 technique was also successfully applied to eliminate *Mitogen-activated protein kinases* (*SlMAPK3*). Through these studies, it was possible to obtain information on the regulatory mechanism of *SlMAPK3*-mediated drought tolerance in tomatoes [102,103]. The identification of *GmMYB118* led to the finding that *GmMYB118* improves drought tolerance and salinity in soybean and Arabidopsis [104], while *GmMYB114* improves drought tolerance in soybean [105]. An abscisic acid receptor (ABA) was investigated in wheat. Studies, such as the one by Mao et al., 2021, have shown that TaPYL1-1B acts to improve wheat drought tolerance and grain yield [106].

The annexin genes (*OsANN3* and *OsANN5*) were knocked out by CRISPR/Cas9 in rice, demonstrating their role in cold resistance [107,108]. The gene knockout, created by CRISPR/Cas9, *9-cis-epoxycarotenoid dioxygenase* (*NCED3*), showed considerable tolerance to multiple stresses in rice. Lou et al. [109] and Lou et al. [110] showed that osmotic *stress/ABA-activated protein kinases* (*SAPK1* and *SAPK2*) contribute to salinity tolerance in rice.

*NAC* is one of the largest families of transcription factors (TFs) that act in the regulation of responses against various plant stresses, and many of these TFs play an essential role in stress tolerance [111]. Studies with soybean show that *GmNAC06* and *GmNAC8* play an important role in tolerance to salt and drought stress, respectively [111,112]. In rice, *OsNAC006*, *OsNAC14,* and *OsNAC45* contribute to drought and heat tolerance, drought tolerance, and salinity tolerance, respectively [113,114,115]. Other examples of CRISPR/Cas9-driven TFs include ARF4, used to improve salinity tolerance and osmotic stress in tomato [116]; *GmAITR*, used to increase salinity tolerance in soybean [117]; and six *AITR* genes (1, 2, 3, 4, 5 and 6) in Arabidopsis, which resulted in greater tolerance to drought and salt [118].

Genes sensitive to abiotic stresses were also reported in the articles selected for this SR. Shi et al. [26] increased the levels of *ARGOS8* transcripts and used CRISPR-mediated HDR to integrate the *GOS2* promoter in the region upstream of *ARGOS8*, thus obtaining greater drought tolerance and higher grain yields. The *nonexpressor of pathogenesis-related gene 1* (*SlNPR1*) was knocked out by CRISPR/Cas9 in tomato, which resulted in drought tolerance, and drought-related genes were downregulated, confirming the regulation of *SlNPR1* in response to water deficit [78]. With the knockout of the *stearic acid gene* (*PtSAD*), it was possible to determine that it acts as a negative regulator in the heat response in *Pinellia ternata*, and its knockout is a potential bioengineering strategy to overcome the negative effects of heat in summer periods [119]. Knockout of a *catabolic ABA* gene (*OsABA8ox2*) strongly improved drought tolerance in rice, while overexpression in seedlings made them hypersensitive to drought, suggesting that *OsABA8ox2* contributes to the drought response in rice [120]. The cell wall/vacuolar inhibitor gene (*C/VIF1*) negatively regulates salt tolerance in *Arabidopsis*, affecting the response to abscisic acid (ABA) [121].

Wang et al. [122] knocked out the transcription factor phytochrome interaction factor4 (*SlPIF4*) and thereby verified increased susceptibility to cold, while the overexpression of SlPIF4 increased cold tolerance in tomato plants. Pan et al. [123] also generated SlPIF4 knockout mutants through CRISPR/Cas9, which showed that anthers had higher tolerance to cold due to a reduced sensitivity to tapetal temperature, while overexpression of *SlPIF4* conferred pollen abortion, delaying tapetal-programmed cell death. *PdGNC*, a member of the GATA transcription factor family, was studied in Populus. The mutant knockout exhibited increased stomatal opening and water loss with reduced drought tolerance. Thus, it was possible to understand that *PdGNC* activates *PdHXK1* (a key gene of hexokinase synthesis), resulting in a noticeable increase in hexokinase activity in poplars submitted to water deficit which has consequences in tolerance increments [124].

Wang et al. [125] and Ye et al. [126] attempted to reduce mineral toxicity using CRISPR/Cas9 technology and reported significant results in their studies. The mutant rice plant, with the knockout of the genes *OsARM1* (*Arsenite-responsive MYB1*) and *OsPT4* (*P transporter genes*), generated by CRISPR/Cas9 showed tolerance to arsenic. Epigenetic modification approaches and a CRISPRa dCas9HAT system with histone acetyltransferase (*HAT*), which can upregulate the activity of a targeted promoter, was constructed by Paixão et al. [127]. This application demonstrated tolerance to water stress through the positive regulation of *ABA-responsive element binding protein 1*/*ABRE binding factor* (*AREB1*).

*miRNAs* belong to a class of small non-coding RNAs that directly regulate the functions of specific messenger RNAs through transcriptional or translational repression [128]. In addition to the target genes, *miRNAs* can also be used for gene editing. Several studies highlight the drought tolerance and salinity responses of *miRNAs* in rice, *Arabidopsis*, and soybean [125,128,129,130,131].

Gene editing aiming to broaden the knowledge of genes and transcription factors involved in the signaling of abscisic and jasmonic acids has been the target of efforts to increase tolerance to different abiotic stresses, such as heat, cold, drought, and the concentration of heavy metals, among others. Biosynthesis of ABA (15) and biosynthesis of jasmonic acid (3-oxo-2-2′-*cis*-pentenyl-cyclopentane-1-acetic, JA) (2) were among the most reported metabolic pathways in the articles evaluated (Figure 8). ABA is a miniscule molecule classified as a sesquiterpene [132]. It is one of the five characteristic phytohormones that helps control many plants’ developmental and growth characteristics, such as leaf abscission, inhibition of fruit ripening, and biotic and abiotic stress [133]. The importance of ABA biosynthesis in tolerance to abiotic stress is reported in several studies [83,132,134,135,136,137].

JA is an endogenous growth-regulating substance found in higher plants [138]. It is a plant signaling molecule related to plant defense mechanisms. It induces the expression of genes encoding specific proteins, such as protease inhibitors, enzymes involved in the production of flavonoids, and different proteins related to diseases, and it also plays an important role in the defense of plants against damage [139]. In the selected studies, JA is a compound that plays a key role in the response to abiotic stress, including drought and cold tolerance [139,140,141].

### 3.5. Methods for Editing with CRISPR/Cas

Among the protocols cited, the most used were those proposed by Ma et al. [47], who reported a CRISPR/Cas9 plant vector system that allows the efficient editing of several genes in monocotyledonous and dicotyledonous plants, enabling the assembly of several gRNA expression cassettes in a single binary vector CRISPR/Cas9 by Golden Gate or Gibson Assembly. Xing et al. [48] developed a set of CRISPR/Cas9 binary vectors based on the pGreen or pCAMBIA backbone, as well as a set of gRNA module vectors, as a toolkit for multiplex genome editing in plants.

The pCAMBIA vectors and their derivatives are the most commonly used binary vectors for a variety of plant species [82,112,114,141,142,143]. The *Agrobacterium* transformation system is still a widely used methodology in several plant species. The main reasons for using this system are its transformation efficiency, low operational cost, and the simplicity of the transformation and selection protocols [144]. Although *Agrobacterium*-mediated delivery is very efficient, it also has some disadvantages, such as the possible random integration of plasmid sequences into the host genome [30]. In addition, this methodology needs to adapt to the current regulations of genetically modified organisms (GMOs) based on processes, thus hindering the commercialization of improved varieties [25].

To address these challenges, attempts were made to deliver RNPs synthesizing small guide RNAs and the Cas9 protein directly into plant cells, with the advantage of obtaining mutants without the presence of exogenous DNA, thus reducing the effects of off-targets and eliminating traces of foreign DNA elements [145]. In the selected articles, RNPs were delivered by means of polyethylene glycol (PEG) in chickpea protoplasts [25] and directly delivered into the potato apical meristem by biobalistic [146].

Cas9 protein type II class II, developed from *S. pyogenes*, was the most used in the articles of this SR. Cas9 nucleases are guided by CRISPR RNAs (crRNAs), which resemble trans-activating crRNAs (tracrRNAs) and facilitate the formation of the ribonucleoprotein complex [147,148]. However, most Cas9 genome editing applications use gRNA, which is designed by fusing crRNA and tracrRNA into a single RNA molecule.

Generally, CRISPR/Cas9 requires a target site of 17 to 20 base pairs (bp) directly adjacent to a 5’-NGG PAM sequence (protospacer adjacent motif) to be effectively recognized by gRNA. In the selected studies, several gRNA were designed with different target sequences to direct Cas9 to specific corresponding sites [149,150,151,152]. According to Ma et al. [47], this is an important resource because the ability of Cas9 to edit several loci simultaneously in the same individual has many potential applications in basic and applied research, such as the mutation of several members of gene families or functionally related genes that control complex characteristics.

In addition to Cas9, the only other nuclease identified in this review was Cas12a, used for knockout of the SlHKT1;2 allele and related to salinity tolerance in tomatoes [153]. CRISPR/Cas12a is a class II endonuclease type V that was developed from *Prevotella* and *Francisella* [147,154]. Compared to Cas9, which requires a crRNA and a tracrRNA complex, Cas12a requires a gRNA complex (crRNA). Compared to Cas9, Cas12a generates cohesive ends, which increases the efficiency of the insertion of a desired DNA fragment at the site cleaved by Cas12a using complementary DNA ends through a mechanism known as HDR, which produces blind extremities [155]. In addition, Cas12a recognizes a PAM region rich in T 5’-TTTN-3 ‘compared to the G-rich PAM sequence, NGG, in Cas9 [153].

Different forecasting tools were used to identify the occurrence of possible off-target effects. Some studies have used online forecasting tools to predict possible off-target effects (Appendix A). Off-target effects are defined as unintentional cleavage and mutations at non-directed genomic sites in a similar but not identical sequence compared to the target site [38].

Among the methods used for CRISPR/Cas mutation detection, sequencing was the most commonly used (81), followed by PCR (70). Genome sequencing is a sensitive and robust method that can detect off-target effects by whole genome sequencing in gene knockout cells [31]. However, Modrzejewski et al. [38] highlighted that these detection methods may be biased given that first, potential off-target sites are predicted using bioinformatics programs, and second, the possible off-target sequences identified are analyzed only for undesired mutations (effects outside of the target), generally ignoring mutations in other loci in the plant genome [37]. To draw any conclusions about the occurrence of off-target effects, a more in-depth analysis is needed; for example, through a SR [38].

### 3.6. Auxiliary Methods

The auxiliary methods to the CRISPR/Cas technique most cited in this SR were RT-qPCR (109) and transgenic or cisgenic methods (67) (Figure 10). The identification of target genes related to the trait in which one wishes to edit is the fundamental part of the gene editing process [145]. Gene expression patterns reflect the trend of gene activity and provide information on the gene function and gene regulation networks in plants under abiotic stresses [156]. One of the most commonly used methods to evaluate gene expression is RT-qPCR, considered a reliable, sensitive, and accurate technique [157].

Transgenic or cisgenic methods were used in the articles selected to explore the function of genes through overexpression. Through this strategy, the authors confirmed that overexpression of genes such as *OsNCED3* (*9-cis-epoxycarotenoid dioxygenase 3*), *SlHyPRP1* (*protein domains of tomato hybrid proline-rich protein 1*), *GmNHX5* (*sodium/hydrogen exchanger* gene), and *OsPUB67* (*U-Box E3 ubiquitin ligase*), among others, leads to tolerance to abiotic stresses, confirming that they are genes that act as positive regulators for tolerance to multiple stresses [23,79,132,143]. The overexpression of *OsABA8ox2* ((ABA) 8′-hydroxylase), *OsProDH* (*proline dehydrogenase*), *OsRR9*, and *OsRR10* (*type-A response regulators*) led to greater sensitivity to drought, heat, and salinity, respectively [120,158,159].

The RNA-seq technique has been used by several authors. The analyses allow the identification and quantification of the relative expression levels of some positively or negatively regulated genes under different abiotic stress conditions [93,131,160,161].

### 3.7. Risk of Bias

Biases are defined as systematic errors in scientific studies that cause distortions in the results, compromising the internal validity of these studies. It is not possible to state with certainty whether a study is biased or not, but we can evaluate the risk of bias of the studies through a careful evaluation of its methodological quality [46]. There are several tools available to assess risk of bias (Robins, Cochrane, and Quadas-2, among others), and they are of great value in health studies and with related questions. For this reason, an adaptation of the Cochrane protocol was conducted [162] to evaluate the methodological quality of our SR, creating three questions that guide studies on CRISPR/Cas technology in plants: 1. Was off-target activity investigated? 2. Was a phenotypic analysis performed? and 3. Is the identified protein studied? These are essential questions that confirm whether editing using CRISPR/Cas was efficient, reaching the target site or not. Through these questions, it was possible to evaluate these articles, most of which were classified as having a low risk of bias and good methodological quality since they included these three essential analyses.

### 3.8. Final Considerations, Limitations, and Future Perspectives

Rapid global climate changes are expected to contribute to the abiotic and biotic stress conditions plants experience and will consequently challenge the food and nutritional security of the global population. The CRISPR/nuclease model (Cas9 or Cas12a) has emerged as a promising method for gene editing in plants due to its ease of use and multifaceted applicability.

In this study, several studies were conducted with CRISPR to edit genes that confer tolerance to abiotic factors such as water deficits, salinity, temperature extremes, and metal toxicity in the last seven years. Plant responses to stress factors are mediated by several complex-signaling pathways and knockouts using CRISPR will accelerate the characterization of the genes and transcription factors related to multiple abiotic stresses, thus enabling the collaborative network between genes to be deciphered more quickly.

Research on genes related to abiotic stresses is far from being complete due to the complexity of the regulatory network, and perhaps, for this reason, the articles mainly used gene knockout technology by CRISPR/Cas; however, a limited number of studies considered systems of precision editing. Tolerance to multiple abiotic stresses is a quantitative characteristic; therefore, the simultaneous editing of several genes will achieve better results in plant genetic improvement.

Regarding the methods used for editing, several gRNAs were designed with different target sequences to direct Cas9 to specific corresponding sites; however, adequate care should be taken when designing gRNAs since off-targets are a major limitation. Among the methods used for mutation detection, sequencing is a sensitive and robust method that can detect off-targets. The regeneration of explants in most cereals is still a challenge because it is laborious and represents a limitation in CRISPR/Cas-based gene editing. In addition to Cas9, Cas12a is a new tool for efficient genome editing, including editing without exogenous DNA in plants, with greater efficiency, specificity, and potentially broader applications than those of CRISPR/Cas9.

Through CRISPR/Cas, it is possible to edit and introduce alleles for positive traits in elite cultivars; however, advances in discussions on the acceptability of GMO products will still be necessary, and at the same time, RNPs are a DNA-free strategy for editing a genome, considering that the generated products will be free of exogenous DNA.

All findings cited in this SR were based on articles with methodological quality confirmed by a risk of bias analysis, which determined that most of the studies included had a low risk of bias. Among the most studied crops, rice, the *A. thaliana* model plant, and tomato stand out, with their number of studies becoming concentrated starting in 2021. This data suggests that the use of this technique in genetic breeding, as to abiotic factors is moving from the test phase and entering the practical phase regarding agricultural crops, although most studies still consider the model plant *A. thaliana*. No scientific studies of gene editing for tolerance to abiotic stresses by CRISPR/Cas were found in important crops, such as sugarcane, banana, and cassava, which are among the most produced food crops in the world. Future studies may provide more up-to-date data on the advances of CRISPR/Cas in terms of gene editing for tolerance to multiple abiotic stresses in these and other crops, as well as provide information on whether there are already plants available for farmers with these modified characteristics.

Currently, a tomato rich in gamma-aminobutyric acid (GABA) edited by CRISPR/Cas9 was the first product to be commercialized in the world for human consumption [163], developed by the company Sanatech Seed in partnership with the University of Tsukuba in Japan.

## 4. Materials and Methods

To prepare for the SR, the State of the Art through Systematic Review (*Start*) software version Beta 3.0.3 (http://lapes.dc.ufscar.br/tools/start_tool, accessed on 9 December 2022) was used. The tool developed by the Federal University of São Carlos (Universidade Federal de São Carlos—UFSCar) aims to support researchers in all stages of SR preparation. Start was developed to improve the validation of the research process, thus reducing risks due to biased information (*bias*).

All the procedures adopted by *Start* followed the Preferred Reporting Items for Systematic Reviews and Meta-Analyses (PRISMA) guidelines (https://doi.org/10.5281/zenodo.7413645, accessed on 9 December 2022). PRISMA allows SRs to be more transparent, with information on why the review was conducted, the approach of the authors, and the information the authors found (results). Thus, the SR process using *Start* occurred in three stages: planning, execution, and summarization.

### 4.1. Planning

After the identification of the need for the SR, the protocol was developed (https://doi.org/10.5281/zenodo.6807191, accessed on 9 December 2022) adopting an edition provided by the *Start* software which contains the following information: article title, authors, objective, keywords, research questions, research sources, inclusion/exclusion criteria, and definition of the type of study. The main topic of research which guided the systematic review was: How may CRISPR data regarding gene edition for tolerance to abiotic factors in the last seven years contribute to genetic breeding in plants? From there on, the secondary questions were defined and are presented in Table 2.

### 4.2. Execution

Following the protocols established for the preparation of the SR, the search string was based on the five inclusion components of population, intervention, comparison, outcome, and study type (PICOS) [164] (Table 3). The use of PICOS improves the specificity and clarity of the problems, promoting more complex search strategies, which allows for more accurate results during the search by also reducing bias-prone errors. A string was also created considering model plants for biological studies and the 20 most important agricultural crops from 2020 data from the Food and Agriculture Organization (FAO).

In total, six electronic databases were used: Google Scholar (https://scholar.google.com.br/?hl=pt, accessed on 9 December 2022), Springer (https://link.springer.com/, accessed on 9 December 2022), CAPES Journal Portal (https://www-periodicos-capes-gov-br.ez103.periodicos.capes.gov.br/index.php?, accessed on 9 December 2022), Web of Science (https://www-webofscience.ez103.periodicos.capes.gov.br/wos/woscc/basic-search, accessed on 9 December 2022), PubMed Central (https://www.ncbi.nlm.nih.gov/pmc/, accessed on 9 December 2022) and CABI Direct (https://www-cabdirect.ez103.periodicos.capes.gov.br/, accessed on 9 December 2022). The searches were performed between April 2021 and 3 March 2022. Some databases limit the number of articles extracted to 1000, such as Google Scholar, where the search string identified 5080 articles; however, it was only possible to extract 990 articles after classification by relevance. The results of the searches in each database were imported into the BIBTEX, MEDILINE, or RIS formats, compatible with *Start*. Relevant articles published after the beginning of the selected work were included manually.

The search strategy for the databases is documented in Table 4. One string was developed for the databases Google Scholar, Springer, CAPES Journal Portal, and CABI Direct, while another was developed only for the Web of Science because of the different search preferences in this particular database. In both cases, the search string was designed to include a wide range of articles related to gene editing by CRISPR/Cas for plant tolerance to abiotic stresses.

Subsequently, the selection and extraction steps were performed. In these phases, the predetermined inclusion criteria (I) were as follows: (I) articles that answered the protocol questions (Table 2), and exclusion criteria (E): (E) theses, dissertations, manuals; (E) book chapter; (E) articles not written in English; (E) articles that did not use CRISPR/Cas to investigate abiotic factors; and (E) articles that did not address the topic.

### 4.3. Summarization

This step included the development of graphs, bibliometric maps, tables, and word clouds to compose the SR.

### 4.4. Risk of Bias Analysis

To evaluate the risk of bias in individual studies, an adaptation of the Cochrane risk of bias tool protocol was performed [162]. Two authors (FSN and MSM) evaluated the methodological quality of the included studies. The instrument for assessing the risk of bias included three questions:Was off-target activity investigated?Was a phenotypic analysis performed?Is the identified protein studied?

When considering all questions, the risk of bias was categorized as “high” when the study had as much as 33.39% of the score as “yes”; “moderate” when the study had as much as 66.6% of the score as “yes”; and “low” when the study had more than 66.6% of the score as “yes”.

### 4.5. Systematic Data Analysis

The frequency of articles for each answer of each research question described in Table 2 was calculated. Thus, the data were expressed as the number of articles per response. The graphs were prepared in the statistical program R [165] with the *ggplot2*, *reshape2*, and *ggpubr* packages. A bibliometric analysis was included for the data collected, and bibliometric maps were generated in the VOSviewer_1.6.17 program [166].

## Figures and Tables

**Figure 1 plants-12-00305-f001:**
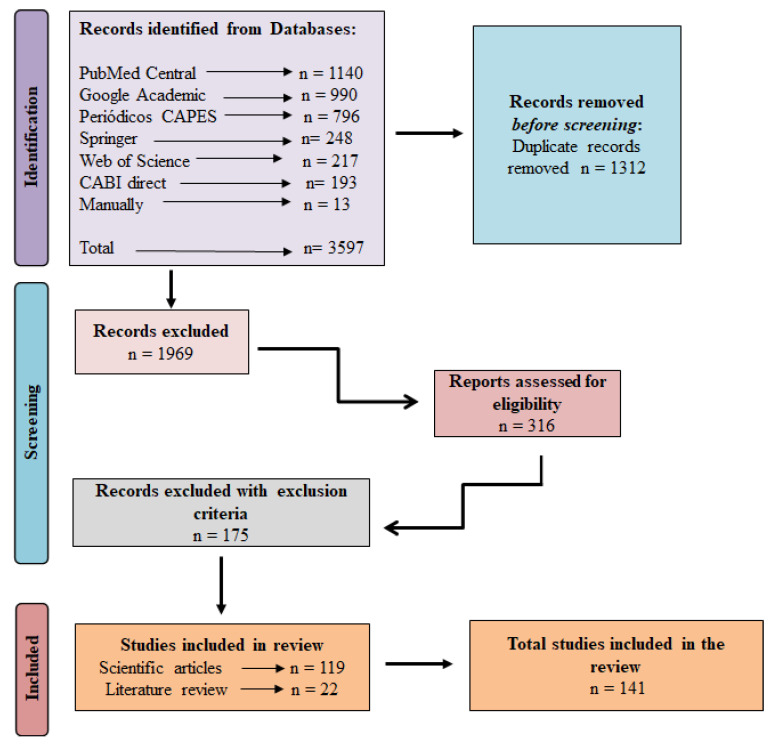
PRISMA flow diagram of the studies included in the systematic review of CRISPR/Cas technology used to edit genes for tolerance to abiotic stresses in plants.

**Figure 2 plants-12-00305-f002:**
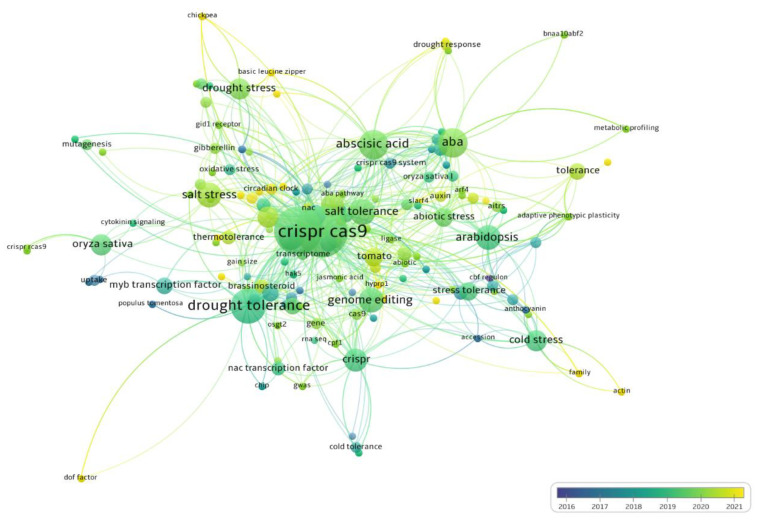
Bibliometric map of the keywords in the selected articles on CRISPR/Cas technology used to edit genes for tolerance to abiotic stresses in plants during the extraction phase of this systematic review.

**Figure 3 plants-12-00305-f003:**
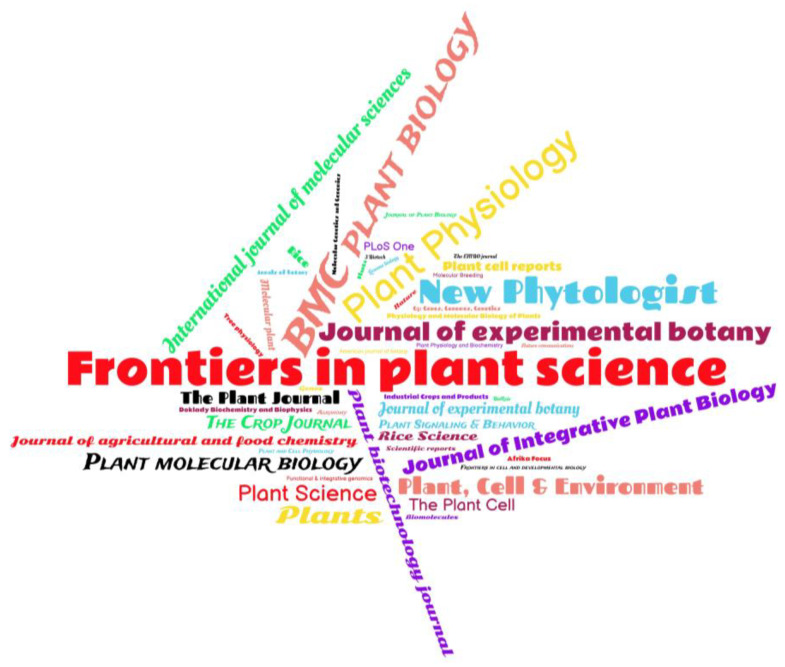
Word cloud of the journals that published the most on CRISPR/Cas being used to edit genes for tolerance to abiotic stresses in plants in the last seven years.

**Figure 4 plants-12-00305-f004:**
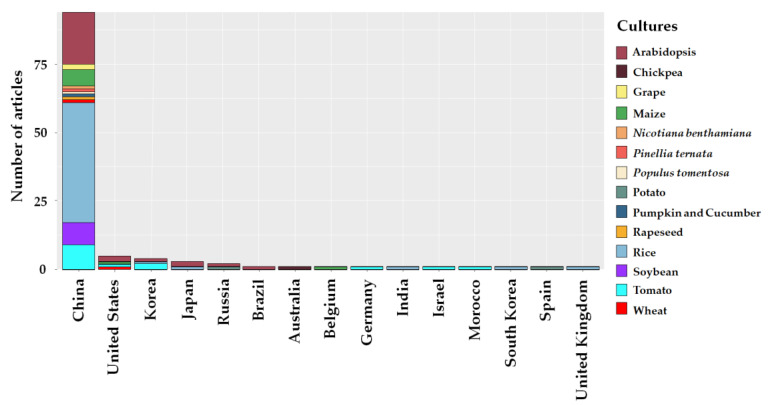
Main countries that have studied CRISPR/Cas being used to edit abiotic stress tolerance genes in the last seven years.

**Figure 5 plants-12-00305-f005:**
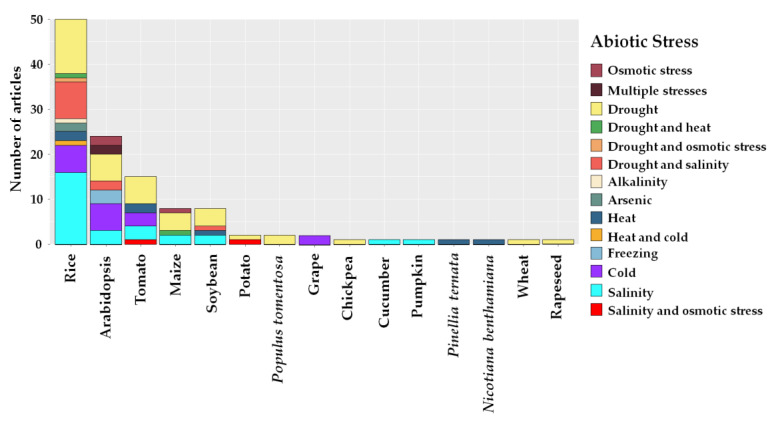
Frequency of crops and abiotic stresses most reported in studies with CRISPR/Cas conducted in the last seven years. Article frequency considered that more than one crop and stress was studied per article.

**Figure 6 plants-12-00305-f006:**
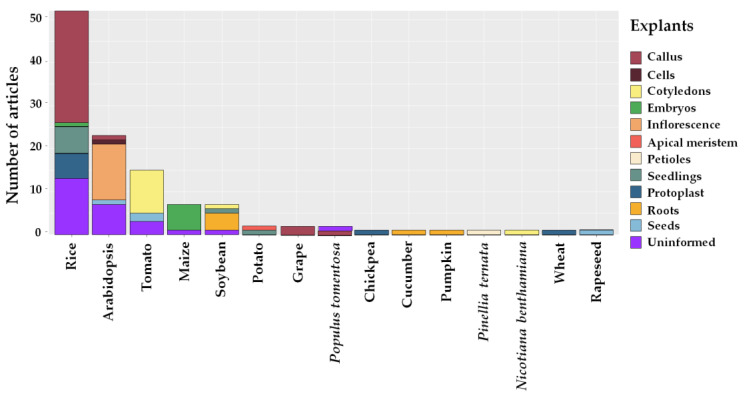
Explants used in studies on gene editing with CRISPR/Cas for tolerance to abiotic stresses in plants in the last seven years.

**Figure 7 plants-12-00305-f007:**
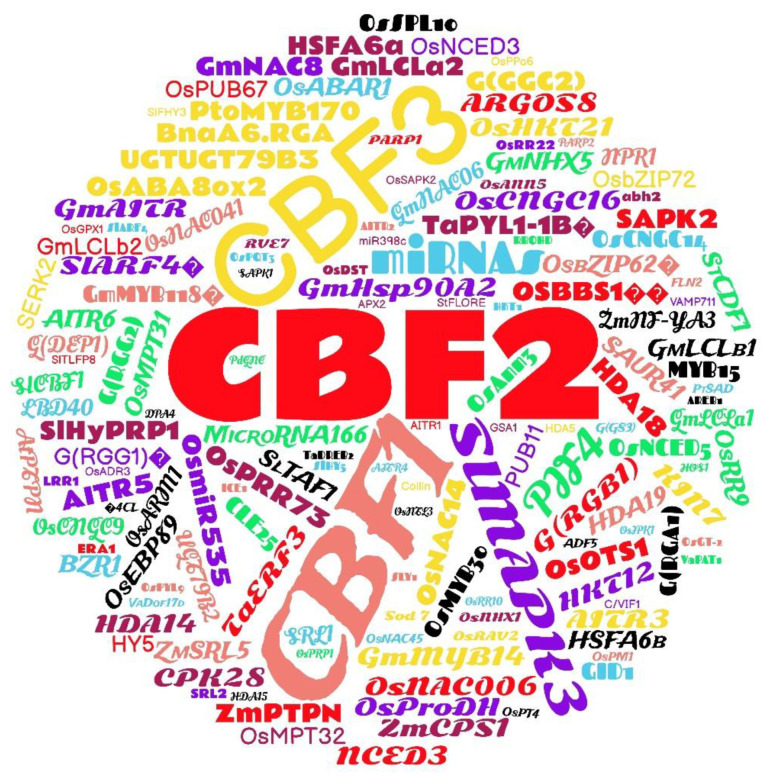
Word cloud of genes related to abiotic stresses identified in studies that used CRISPR/Cas gene-editing technology in the last seven years.

**Figure 8 plants-12-00305-f008:**
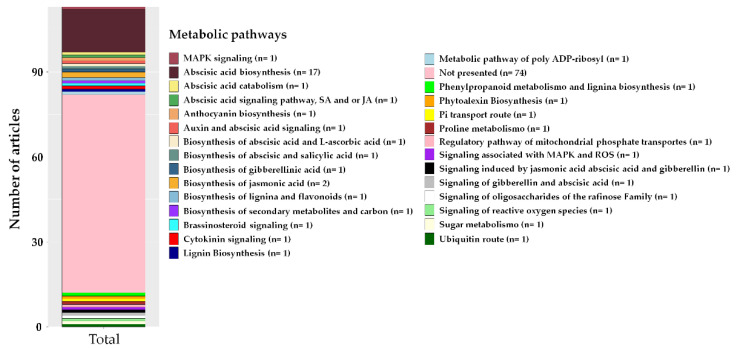
Metabolic pathways reported in the articles on CRISPR/Cas-associated gene editing for tolerance to abiotic stresses published in the last seven years, n = number of articles.

**Figure 9 plants-12-00305-f009:**
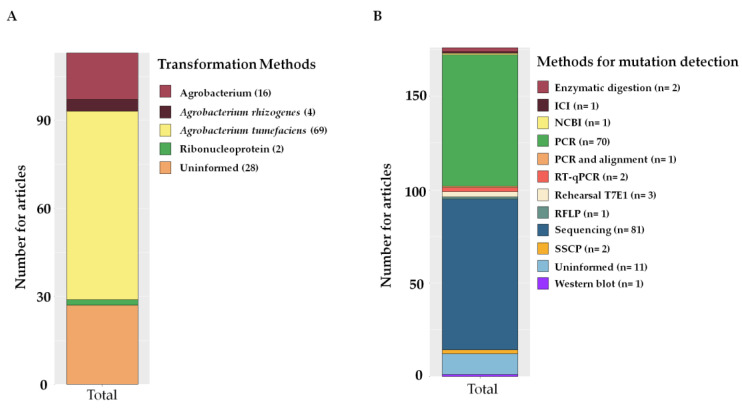
Graph of stacked bars represent the frequency of articles with different delivery methods and mutation detection with CRISPR/Cas. (**A**) Main delivery methods for genes of interest related to abiotic factors used in editing by CRISPR/Cas in the last seven years. (**B**) Methods used to detect mutations generated by CRISPR/Cas identified in articles from the last seven years. Determining the frequency involved considering that more than one method for mutation detection was used per article, n = number of articles.

**Figure 10 plants-12-00305-f010:**
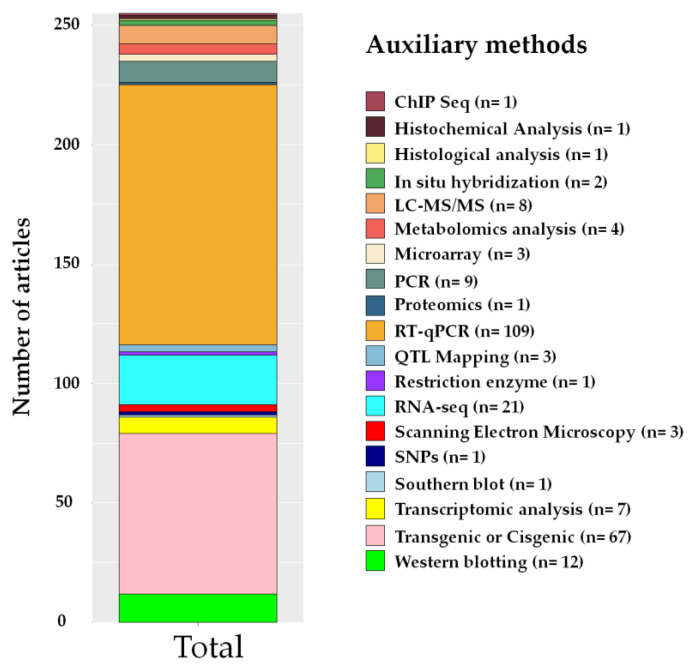
Auxiliary tools used with the CRISPR/Cas technique for comparison between knockout with the control and/or with the overexpression of mutants identified in articles on tolerance to abiotic stresses in the last seven years. The frequency was determined considering that more than one auxiliary method was used per article, n = number of articles.

**Figure 11 plants-12-00305-f011:**
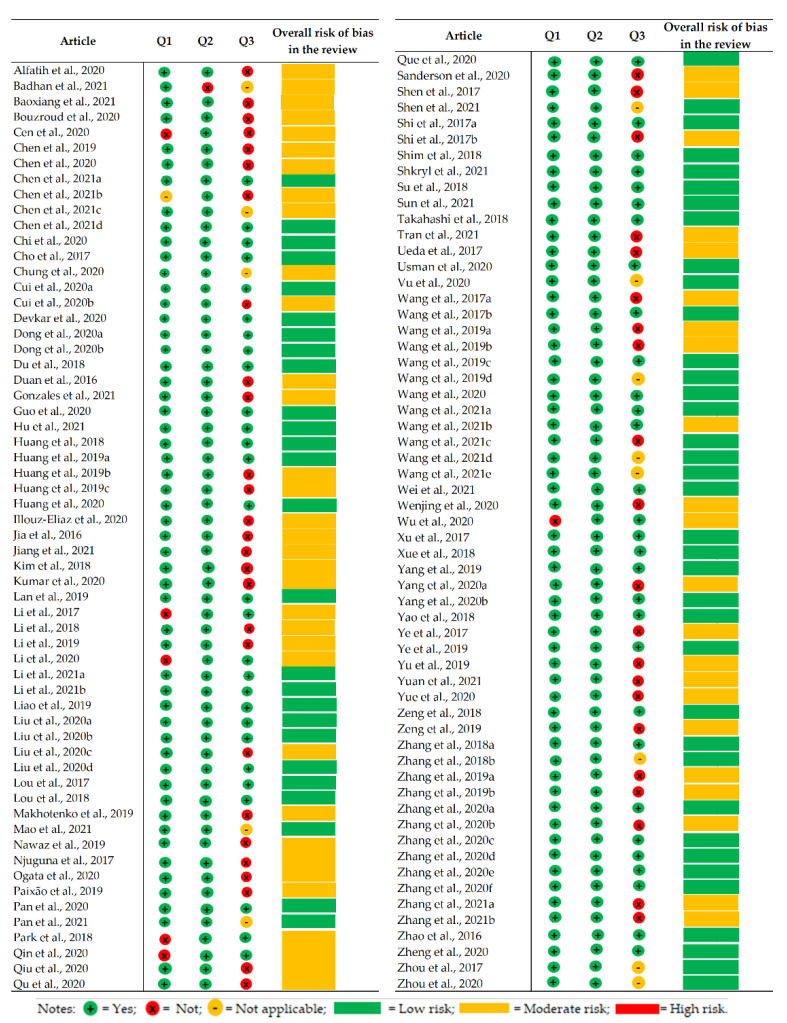
Summary of the risk of bias assessment in the articles on CRISPR/Cas technology used for gene editing for tolerance to abiotic stresses published in the last seven years.

**Table 1 plants-12-00305-t001:** Literature reviews published in the last seven years on CRISPR/Cas technology applied to plant breeding with emphasis on abiotic stresses.

Article	Culture	Objective of the Review
[49]	Cultures in general	Applications of CRISPR/Cas9-mediated gene editing to produce plants grown under stressful environmental conditions.
[50]	Cultures in general	Development of crops with high yields and tolerance to abiotic stresses.
[51]	Rice	CRISPR/Cas9 to develop heat-tolerant rice and tolerance to water deficits and floods.
[33]	Cultures in general	CRISPR/Cas9 for tolerance to abiotic stress.
[19]	Cultures in general	Comprehensive overview of CRISPR/Cas technology to improve tolerance to abiotic stress.
[52]	Cultures in general	CRISPR/Cas9 and *ERFs* * used for tolerance to abiotic stress.
[31]	Rice	Rice generated to be capable of sustaining growth under conditions of high salinity, using CRISPR/Cas.
[53]	Cultures in general	Application of CRISPR/Cas technology for the elimination/deactivation of genes associated with abiotic stresses.
[12]	Cultures in general	Biotic and abiotic factors.
[32]	Cultures in general	Application of CRISPR/Cas for tolerance to abiotic stress.
[54]	Cultures in general	CRISPR/Cas9 to understand tolerance to abiotic stress.
[55]	Cultures in general	Tolerance to water stress.
[56]	*Arabidopsis*	Functional role of *CNGC19* and *CNGC20* * in *Arabidopsis* using CRISPR/Cas9.
[57]	Cultures in general	Genome editing approaches based on CRISPR/Cas that have been used in plants for tolerance to abiotic stress.
[8]	Cultures in general	CRISPR/Cas approaches and their efficiency to improve plant growth and responses to abiotic stress.
[58]	Cultures in general	Applications of omics and CRISPR/Cas9 for the development of stress-tolerant cultures.
[59]	Cultures in general	Genome editing based on CRISPR/Cas9 in targeting *HyPRPs** for tolerance to multiple stresses.
[60]	Cultures in general	Tolerance to drought, yield, and domestication.
[61]	Cultures in general	Abiotic and biotic factors.
[30]	Banana	Recent and prospective advances in the application of genetic modification and genome editing for the development of bananas resistant to high temperatures and water deficits.
[62]	Cultures in general	Evaluation of available tools and target genes to obtain plants with greater tolerances to abiotic stresses.
[29]	Cultivation plants	Production of multiple stress-tolerant crops using CRISPR/Cas9.

* Ethylene Response Factor (ERF); Cyclic Nucleotide-gated channel (CNGC); Proline Rich Proteins (HyPRPs).

**Table 2 plants-12-00305-t002:** Questions about using CRISPR/Cas technology to edit genes for tolerance to abiotic stresses in a systematic review of studies published in the last seven years.

Questions
Q1. Which cultures have been edited using the CRISPR/Cas technique?
Q2. Which genes have been edited using the CRISPR/Cas technique?
Q3. What metabolic pathways are reported in studies with CRISPR/Cas?
Q4. Which countries or continents most widely use the CRISPR/Cas technique for tolerance to abiotic stresses?
Q5. Which enzymes other than Cas9 are used in CRISPR?
Q6. What protocols are proposed for editing with CRISPR/Cas?
Q7. Which explants are most used for gene editing with CRISPR/Cas?
Q8. What type of vector and bacteria are most reported as being used with CRISPR/Cas?
Q9. What methods are used to confirm the efficiency of the CRISPR/Cas technique?
Q10. Which abiotic factors is CRISPR/Cas used to modify?
Q11. What auxiliary methods to CRISPR/Cas for tolerance to abiotic stresses are used?

**Table 3 plants-12-00305-t003:** Definition of the PICOS terms for the research “question” used in the systematic review of CRISPR/Cas technology being used to edit genes for tolerance to abiotic stresses in plants published in the last seven years.

Description	Abbreviation	Components of the Question
Population	P	Agricultural crops with abiotic stresses.
Interest/Intervention	I	Gene editing based on CRISPR/Cas technology for plant breeding.
Comparison	C	Methods of plant breeding that do not include editing genes with CRISPR/Cas.
Outcome	O	Editing genes that confer tolerance to abiotic stresses in plants.
Study type	S	Scientific articles and literature reviews.

**Table 4 plants-12-00305-t004:** Search keywords used for the systematic review of CRISPR/Cas technology in the editing of genes for tolerance to abiotic stresses in plants of the last seven years.

Database	Keyword Variations
Google Scholar	(“abiotic factors” OR “water deficit” OR “drought tolerance” OR “salinity tolerance” OR “cold tolerance” OR “heat tolerance”) AND (“CRISPR/Cas9” OR CRISPR-Cas9 OR “CRISPR-Cas in plants”).
Springer	
CAPES Journal Portal	
CABI Direct	
Web of Science	(crop OR crops OR plant OR plants OR seed OR seeds OR Arabidopsis OR Tobacco OR Nicotiana OR “zea mays” OR maize OR wheat OR Triticum OR barley OR hordeum OR rice OR oryza OR soybean OR “Glycine max” OR potato OR Solanum OR “sweet potato” OR “Ipomoea batatas” OR “sugar beet” OR “sugar-beet” OR “fodder beet” OR “beta vulgaris” OR tomato OR cucumber OR cucumis OR onion OR allium OR apple OR apples OR malus OR orange OR “Citrus sinensis” OR banana OR musa OR manihot OR cassava OR “Manihot esculenta” OR sugarcane OR “Saccharum officinarum” OR cotton OR “Gossypium hirsutum” “oil palm” OR “Elaeis guineensis” OR watermelon OR Citrullus) AND (“genome edit*” OR “genome-edit*” OR “genome editing in plants” OR CRISPR/Cas9 OR CRISPR-Cas9 OR “CRISPR/Cas9-targeted mutagenesis” OR “targeted mutagenesis” OR “genome editing technology”) AND (“abiotic factors” OR “Abiotic stress” OR “water deficit” OR “drought tolerance” OR “salinity tolerance” OR “cold tolerance” OR “heat tolerance”)

## Data Availability

Not applicable.

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
