# Peer review of "Gene Editing for Plant Resistance to Abiotic Factors: A Systematic Review"

_plants, 2023, doi:10.3390/plants12020305_

Round 1

Reviewer 1 Report (Previous Reviewer 1)

The authors have responded all my concerns. It can be accepted now.

Author Response

Dear reviewer! We appreciate your important contributions. Best regards and Happy 2023

Reviewer 2 Report (Previous Reviewer 3)

The authors have improved the content and addressed most of my comments in their revised MS, but they need to consider the following aspects. They should not be casual during revising the draft, which may consume more time and delay the reviewing process.

1. My previous comment- 7 (Line 76- non-homologous end junction or joining? I think NHEJ comprises the word- joining)

It is not yet resolved in the revised MS. NHEJ stands for non-homologous end joining. It is still mentioned as - a non-homologous junction! It is a standard scientific term, and the Authors can not change its full form.

2.  My previous comment-12. (Lines 217 and 226- Authors should clarify what they mean by qRT-PCR or RT-qPCR)

Same issue. It is not yet resolved in the revised MS. The term RT-qPCR should be revised as - “quantitative reverse transcription PCR (RT-qPCR)” on page 9, and then only the abbreviation should be used in the following paragraphs (for example, page 10).

Author Response

We would like to acknowledge your comments and indicate that all have been considered and included in the manuscript.

Comments and Answers

The authors have improved the content and addressed most of my comments in their revised MS, but they need to consider the following aspects. They should not be casual during revising the draft, which may consume more time and delay the reviewing process.

  1. My previous comment- 7 (Line 76- non-homologous end junction or joining? I think NHEJ comprises the word- joining)

It is not yet resolved in the revised MS. NHEJ stands for non-homologous end joining. It is still mentioned as - a non-homologous junction! It is a standard scientific term, and the Authors can not change its full form.

A = Adjusted, the term “non-homologous end joining (NHEJ)” was modified on page 2.

  1. My previous comment-12. (Lines 217 and 226- Authors should clarify what they mean by qRT-PCR or RT-qPCR)

Same issue. It is not yet resolved in the revised MS. The term RT-qPCR should be revised as - “quantitative reverse transcription PCR (RT-qPCR)” on page 9, and then only the abbreviation should be used in the following paragraphs (for example, page 10).

A = Adjusted. The term “Quantitative reverse transcription PCR (RT-qPCR)” was added on page 9. 

Reviewer 3 Report (Previous Reviewer 4)

The authors have addressed my concerns and substantially improved their manuscript. I feel the current version is ready for acceptance. 

Author Response

Dear reviewer! We appreciate your important contributions. Best regards and Happy 2023

This manuscript is a resubmission of an earlier submission. The following is a list of the peer review reports and author responses from that submission.

Round 1

Reviewer 1 Report

In this manuscript, authors systematically summarized and analyzed the study of the use of CRISPR in gene editing for tolerance to abiotic stress in plants. I think that the information presented in this manuscript can help to understand the current state of CRISPR in the editing of genes and noncoding sequences involved in the regulation of tolerance to multiple abiotic stresses, which are potential for use in future plant genetic improvements. This manuscript was well written and organized. There are only some minor concerns with me.

1. Figure 2, 3, 7 should be replaced with higher definition ones. And please delete the VOSviewer label on Figure 2 and 3.

2. Line 175, delete the blank space between word “genes’ and “and”.

3. Please correct all gene names in Italic format in your manuscript.

4. Line 417, replace word “cas9” with “Cas9”.

5. Part 3.8, could you also describe if there will be some potential threats to the environment and ecosystem with the gene editing technology?   

Reviewer 2 Report

This review is out of scope of the Plant journal.

Reviewer 3 Report

This manuscript has analyzed published literature about using CRISPR/Cas9-based genome editing tools to design resistant plants for abiotic factors. I have thoroughly read the paper, and the systematic review (SR) is of interest to plant genome editors, and the authors have summarized some new aspects about the topic. Mainly, parameters (questions) defined/formulated by authors to extract the information about CRISPR-based plant genome editing regarding abiotic factors are generally not found in recent reviews.  

I am providing my comments for further improvement of SR as follows.

1.      The authors have not mentioned the types of CRISPR-based tools and mainly wrote their review by focusing on using CRISPR/Cas9 for gene knockout studies. There are abiotic stress-related studies of using other CRISPR-based tools like CRISPRa, epigenetic modification, HDR, engineering of genetic elements other than protein-coding genes (promoter), systematic deletion of functional domain-coding regions rather than gene knockouts, etc., which is summarized in a recent review (https://doi.org/10.1016/j.xplc.2022.100417). Discussion about this aspect in the introduction would help readers.

2.      Cpf1 is renamed Cas12a, and the term Cas12a has been used recently. Authors may consider using it in the whole text.

3.      CRISPR or CRISPR/Cas or CRISPR/cas or CRISPR/Cas9?

4.      Line 66- why the full form of CRISPR is in italic? Also, this sentence needs to be revised for clarity. Although SpCas9 is the most used Cas endonuclease, it is not the only enzyme repurposed for GE.

5.      Line 71-72 – The author may consider updating the sentence with the precise information that- CRISPR tools offer modifications at all layers of central dogma.

6.      gRNA or sgRNA? Please use either of the terms uniformly.

7.      Line 76- non-homologous end junction or joining? I think NHEJ comprises the word- joining.

8.      Line 79- frame displacement mutations should be replaced with frame-shift mutations.

9.      The term- systematic review should be abbreviated when used for the first time. This issue is with several other terminologies used by Authors; therefore, this aspect should be corrected throughout the manuscript.

10.  All the scientific names of plant or bacterial species (e.g., Arabidopsis thaliana, Agrobacterium tumefaciens, Agrobacterium rhizogenes, etc.) and genes (several are in the main text) should be italicized and corrected throughout the manuscript.

11.  Line 202-204 – To introduce the gene of interest into plant cells, the most commonly used delivery methods were via Agrobacterium tumefaciens (63), Agrobacterium rhizogenes (4), Escherichia coli (2), and ribonucleoproteins (RNPs) (2) (Figure 9A).

Escherichia coli was used for CRISPR reagent delivery into plant bodies? Which articles used E. coli for gene delivery into plants? Authors should look into this aspect. Also, authors may carefully look into the other aspects in a manual way to avoid misinterpretation of data due to software limitations.

12.  Line 217 and 226- Authors should clarify what they mean by qRT-PCR or RT-qPCR.

13.  Authors may consider formatting figures with the same font and better resolution.

14.  Also, the use of CRISPR-related terms should be cross-checked for accuracy.

Reviewer 4 Report

This study systematically summarizes and dissects the CRISPR- and abiotic stress-related literature reported since 2015. In this study, Fernanda et al. identified 113 CRISPR- and abiotic stress-related literature and then compared the frequency of reported abiotic stress, journals and countries for publishing and conducting these studies. Next, they further analyze the edited plants, type of explant and transformation, main genes and pathways, type of Cas protein, method for analyzing editing outcome, risk of bias analysis, and literature reviews.

Overall, this work represents a comprehensive review to summarize the advances of CRISPR technology used for the study of abiotic stress. It also provides helpful guidelines and cues for future studies on abiotic stress in plants using CRISPR. To make this review succeed, the key factor is that the authors could collect all related literature. Otherwise, all analyses are performed based on an incomplete database.

Major concerns:

  1. The authors need to further confirm whether they have collected all related literature. To the best of my knowledge, a lot of literature is missing in this study. For example, in tomato,

“Crosstalk of PIF4 and DELLA modulates CBF transcript and hormone homeostasis in cold response in tomato”, “PIF4 negatively modulates cold tolerance in tomato anthers via temperature-dependent regulation of tapetal cell death”, “SlFHY3 and SlHY5 act compliantly to enhance cold tolerance through the integration of myo-inositol and light signaling in tomato”

Therefore, the authors need to re-screen all related literature.

  1. Can authors provide a better method to show Figure 2 and Figure 3? It’s hard to tell the keywords and timeline shown in both figures.
  2. Line 169: can the authors explain how “seeds” could be used as explants for transformation in rice? 

Minor concerns:

  1. Line 92: Provide the full name of “SR” since it is first shown in the text.
  2. Line 74: Please do not mention Fok I here to avoid any misunderstanding.
  3. Figure 4: Re-define the order of countries listed on the x-axis e.g. based on the total number.